# Relationship between health literacy and health-related quality of life in Korean adults with chronic diseases

Inmyung Song 🄾*

College of Nursing and Health, Kongju National University, Gongju, South Korea

* inmyungs@gmail.com

**Data Availability Statement:** The data is owned by a third party and the authors had no special access privileges others would not have. A request for the data used for this study can be made on https://www.khp.re.kr:444/eng/data/data.do. Inquiries

## Abstract

Inadequate health literacy is a risk factor for poor health outcomes and health-related quality of life (HRQoL). So far, the impact of health literacy on HRQoL has been examined for only a few chronic conditions. In this contribution, the relationship between health literacy and HRQoL in Korean adults with chronic conditions is examined using data of the cross-sectional Korea Health Panel Survey from 2021. Health literacy was measured with the 16-item European Health Literacy Survey Questionnaire (HLS-EU-Q16) and HRQoL with the European Quality of Life-5 Dimensions (EQ-5D). Multiple linear regression model was run for the EQ-5D index as the dependent variable. Multiple logistic regression models were implemented for responses to the individual EQ-5D items. 30.8%, 24.6%, and 44.6% of participants had inadequate, marginal, and adequate levels of health literacy, respectively. The EQ-5D index increases with marginal (B = 0.018, $p<0.001$) and adequate literacy (B = 0.017, $p<0.001$) compared to inadequate literacy. People with adequate or marginal literacy were more likely to report no problem with mobility (odds ration [OR] = 1.5; $p<0.001$), self-care (OR = 1.6; $p<0.05$), and usual activities (OR = 1.6 for adequate; OR = 1.4 for marginal; $p<0.01$) than those with inadequate literacy. Adequate health literacy was associated with an increased likelihood of having no problem with anxiety and depression (OR = 1.4; $p<0.05$). In conclusion, inadequate health literacy is prevalent among Korean adults with chronic diseases. Adequate health literacy is associated with better HRQoL and a protective factor for four dimensions of EQ-5D (mobility, self-care, usual activities, and anxiety/ depression).

## Introduction

There is a growing interest in health literacy and its impact on health outcomes [1,2]. Health literacy has been defined as "the degree to which individuals have the capacity to obtain, process, and understand basic health information and services needed to make appropriate health decisions [3]." Over time, the complex construct of health literacy has evolved to represent "the knowledge, motivation and competencies of accessing, understanding, appraising and applying health-related information" in three domains of health care, disease prevention, and

regarding data acquisition should be sent to the Korea Institute for Health and Social Affairs (email: khp@kihasa.re.kr).

**Funding:** The author(s) received no specific funding for this work.

**Competing interests:** The authors have declared that no competing interests exist.

health promotion [4]. However, this critical "capacity" seems to be lacking in a large proportion of the population across countries [5]. According to a systematic review [6], nearly a half of adults in the U.S. have limited health literacy. A third of older adults in England have low functional health literacy, which means that they have difficulties reading and understanding basic written health information [7]. In Korea, nearly one in two adults aged 19 years and over have limited health literacy [8]. This is concerning, provided that Korea is an economically developed country with universal health coverage [9]. Korea is also known as one of highest-achieving countries with respect to scientific literacy [10], but its education is heavily reliant on exam-based education system [11], suggesting health literacy deserves assessment on its own.

Health literacy has been measured with a number of tools including the Rapid Estimate of Adult Literacy in Medicine (REALM) [12], the Test of Functional Health Literacy in Adults (TOFHLA) [6,13], the Newest Vital Sign (NVS) [14], and the Fostering Literacy for Good Health Today (FLIGHT) [15]. One of the newer measures was the European Health Literacy Survey Questionnaire (HLS-EU-Q). HLS-EU-Q was originally developed to measure health literacy in the European population [5], and was translated and used in a number of non-European populations [16]. The original version consisting of 47 items was designed to comprehensively assesses multiple conceptual domains of health literacy in diverse contexts [5], and later was shortened to a 16-item European Health Literacy Survey Questionnaire (HLS-EU-Q16). The efforts to measure health literacy across countries are underway, one example of which is the Health Literacy Study-Asia (HLS-Asia), which is a project organized with a view to provide an overview of the health literacy status in Asia [17].

Limited health literacy is linked to lower use of preventive care [18], greater use of emergency services [12], higher hospitalization rates [19], poorer health outcomes [1,2], and increased mortality [7,20]. As a result, limited health literacy increases economic burden on the health care system as well as on affected individuals [21]. Inadequate health literacy can be particularly problematic for patients with chronic conditions, because taking an active role in their own care is an important component of effective treatment. There is growing evidence that inadequate health literacy is associated with various health outcomes among patients with chronic conditions [22–24]. For example, low health literacy and self-care skills can negatively affect clinical outcomes among patients with cardiovascular disease and diabetes [22]. Low health literacy was associated with poorer control of blood pressure among patients with hypertension [23]. Furthermore, limited health literacy had a negative impact on patient-reported outcomes even after controlling for physician-rated disease activity among patients with systematic lupus erythematosus [24].

Reflecting the subjective experience of the patient, health-related quality of life (HRQoL) is an important patient-reported health outcome in patients with chronic conditions [25,26]. While chronic conditions have a negative impact on HRQoL [27], the adverse impact can be moderated by health literacy because it can improve self-care and self-management [28]. Health literacy is positively associated with HRQoL, measured with the European Quality of Life-5 Dimensions (EQ-5D) index, among spine patients recruited from a single medical center [14]. A chronic condition that garnered particular attention in relation to health literacy is type 2 diabetes [29–31]. Type 2 diabetes patients with adequate health literacy are less likely to experience a decline in EQ-5D over one year than those with low health literacy [30]. While these existing studies offer valuable insights regarding the role of adequate health literacy in HRQoL of patients with chronic disease, they are focused on a specific chronic condition [14,29,30], or based on a small sample of patients from a single [14] and few centers [29,31]. Moreover, the impacts of health literacy on specific dimensions of HRQoL are not fully understood [14].

Therefore, this study aims to evaluate the impact of health literacy on HRQoL among community-dwelling Korean adults with chronic conditions and to determine whether adequate health literacy is associated with increased likelihood of experiencing no problem with each dimension of HRQoL. The following hypotheses were formulated;

1. Health literacy is positively associated with HRQoL in patients with chronic conditions.

2. Health literacy has a differential impact across dimensions of HRQoL.

## Methods

### Data and the study population

This cross-sectional study used data from the Korea Health Panel Survey (KHPS) conducted in a nationally representative large sample of the population between March and July, 2021. The KHPS is a population-based panel survey that started in 2008. The KHPS used a two-stage stratified cluster sampling design [32]. Stratifications were done twice, first based on a large administrative district (city or province) and then on a smaller district. Trained interviewers conducted computer-assisted personal interviews face-to-face at the home of participants using a structured questionnaire.

The 2021 KHPS collected data on health literacy and EQ-5D among adults aged 19 years and older [8]. The KHPS also elicited information on chronic conditions, which included hypertension, diabetes mellitus, chronic hepatitis, alcoholic hepatitis, liver cirrhosis, osteoarthritis of the knee, degenerative arthritis of other joints, rheumatoid arthritis, spinal disc herniation, other spinal disease, stomach cancer, colon cancer, lung cancer, breast cancer, cervical cancer, thyroid cancer, other cancer, angina pectoris, myocardial infarction, cerebral infarction, asthma, emphysema, chronic obstructive pulmonary disease, bronchiectasis, hypothyroidism, hyperthyroidism, depression, dipolar disorder, dementia, chronic renal failure, and other chronic condition.

In the 2021 KHPS, 6,217 households were sampled and 6,190 (99.6%) participated in the survey [8]. A total of 13,530 members of the households responded to the survey. They were asked to indicate if they have any of the aforementioned chronic conditions diagnosed by a physician. In response, a total of 5,865 members reported that they had at least one chronic condition. Among them, 5,663 participants who answered health literacy questions were included for analysis.

### Health literacy

HLS-EU-Q16 is a comprehensive measure of health literacy comprised of 16 questions in three domains: health care (7 items), disease prevention (5 items), and health promotion (4 items). Each item was rated on a 4-point Likert scale (very difficult, difficult, easy, and very easy). 'Very difficult' and 'difficult' were assigned 0, and 'easy' and 'very easy' 1. The scores on all items were summed. The summary scores were then classified into the following three levels of health literacy: inadequate (scores, 0–8), marginal (scores, 9–12), or adequate (scores, 13–16). HLS-EU-Q16 was translated into Korean using the translation and back-translation method [33]. Cronbach's alpha for the Korean version is 0.861. The Korean version was used to assess health literacy in the 2021 KHPS. While the shorter 12-item European Health Literacy Survey Questionnaire (HLS-EU-Q12) was validated in other Asian countries at the time of the 2021 KHPS [34], the nationwide survey relied on the Korean version of the longer HLS-EU-Q16, which had been successfully used in the preceding years.

## Health-related quality of life

HRQoL is a multidimensional construct consisting of physical, psychological, and social functioning dimensions [35]. The 2021 KHPS assessed HRQoL with EQ-5D. EQ-5D was initially developed as a concise measure of HRQoL that could be used across disease areas [36] and has been used to measure HRQoL in patients with chronic conditions [37]. The EQ-5D-3L is a preference-based index measure, based on five dimensions (mobility, self-care, usual activities, pain/discomfort, and anxiety/depression) with three levels of response (no problem, moderate problem, and severe problem). Each dimension of the EQ-5D is addressed by exactly one item. The instrument has been used widely in numerous nationwide surveys in Korea such as the KHPS [8]. Outcomes of interest in this current study were the EQ-5D index and responses to the individual EQ-5D items. The EQ-5D index was obtained using the EQ-5D value set derived from a representative sample of Korean adults [38]. For analysis of responses to the individual EQ-5D items, the three levels of response in each dimension were dichotomized into no problem and problem (inclusive of moderate to severe problems) categories.

## Covariates

Potentially confounding factors for EQ-5D were identified and adjusted for in all regression models. Based on literature review, covariates included socio-demographic variables comprising sex, age, educational level, marital status, employment status, household income quartile and health-related variables such as disability, and multi-morbidity [25,39,40]. These variables could potentially influence EQ-5D and therefore were adjusted for in all regression models to better estimate the relationship between health literacy and EQ-5D. Educational level was classified as no or primary education, middle school, high school, and college and higher. Marital status was divided into currently married, formerly married (divorced/widowed/separated), and single. Employment status was divided into the employed, self-employed, and unemployed. Household income was divided into quartiles. Disability was divided into yes and no. Multi-morbidity was measured as the number of chronic conditions and divided into three categories (one, two, and ≥three conditions).

## Statistical analysis

Sociodemographic and health-related characteristics of the participants were described with the frequency and percentage. The prevalence of health literacy levels across the categories of socio-demographic and health-related characteristics was calculated and the difference was tested by using Rao-Scott chi-square test, which is recommended for analyzing complex survey data [41]. The proportion of the participants reporting problems in EQ-5D dimensions was calculated and compared among health literacy levels. The mean EQ-5D index (±SE) was computed and the difference among health literacy levels was tested using the one-way analysis of variance. A multivariable linear regression model was run to examine if health literacy levels are associated with EQ-5D index. In addition, five multivariable logistic regression models were implemented to assess the relationship between health literacy levels and the likelihood of having no problem in each dimension of EQ-5D. For logistic regression analyses, adjusted odds ratio (OR) and 95% confidence intervals (CI) were calculated.

All statistical analyses were performed by using SAS version 9.4 (Cary, NC, USA). All analyses adjusted for sampling weights that were used in the complex sampling design of the 2021 KHPS. The complex survey uses oversampling of some subgroups of the population and post-stratification adjustments for nonresponses, which cause the issues of unequal selection probability across subgroups and incorrect estimation of standard errors [42]. Sampling weights provided in complex survey data represent the inverse of selection probability. Units

oversampled have a smaller weight value. Weighting was performed using the "surveyreg" and "surveylogistic" procedures in SAS. The Institutional Review Board of Kongju National University approved the study protocol and waived the requirement for informed consent (reference No. KNU_IRB_2023–101).

## Results

In the weighted sample, 55.7% are women, 30.4% in their 60s, 66.6% currently married, and 43.8% unemployed (Table 1). 23.9% have a college-level education. 9.0% of participants have a disability and 22% three or more chronic conditions. 30.8%, 24.6%, and 44.6% have inadequate, marginal, and adequate levels of health literacy, respectively.

The prevalence of health literacy levels varies significantly among the categories of all sociodemographic and health-related variables ($p<0.001$) (Table 2). In particular, adequate health literacy is more frequently observed in men, younger age groups, people with higher education, the single, the employed, higher income quartile, people with no disability, and people with fewer chronic conditions. The proportion of participants reporting problems in EQ-5D

**Table 1. Characteristics of study participants (n = 5,663).**

| Characteristic | Category | N | % (weighted) |
|---|---|---|---|
| Gender | male | 2,400 | 44.3 |
| | female | 3,263 | 55.7 |
| Age, years | 19–49 | 439 | 14.7 |
| | 50–59 | 746 | 22.9 |
| | 60–69 | 1,747 | 30.4 |
| | 70–79 | 1,935 | 20.8 |
| | ≥80 | 796 | 11.3 |
| Educational level | no or primary education | 2,185 | 26.9 |
| | middle school | 1,033 | 15.1 |
| | high school | 1,595 | 34.1 |
| | college or higher | 850 | 23.9 |
| Marital status | currently married | 4,014 | 66.6 |
| | formerly married | 1,467 | 27.5 |
| | single | 182 | 5.9 |
| Employment status | employed | 1,832 | 40.7 |
| | self-employed | 975 | 15.5 |
| | unemployed | 2,856 | 43.8 |
| Household income quartile | 1 (lowest) | 1,413 | 22.0 |
| | 2 | 1,414 | 18.7 |
| | 3 | 1,414 | 22.9 |
| | 4 (highest) | 1,413 | 36.3 |
| Disability | yes | 638 | 9.0 |
| | no | 5,025 | 91.0 |
| No. of chronic conditions | 1 | 2,388 | 51.1 |
| | 2 | 1,689 | 26.8 |
| | ≥3 | 1,586 | 22.0 |
| Health literacy level | inadequate | 2,276 | 30.8 |
| | marginal | 1,445 | 24.6 |
| | adequate | 1,942 | 44.6 |

N is the frequency in the sample. % (weighted) is the population estimate.

**Table 2. Prevalence of health illiteracy levels by sociodemographic and health-related characteristics.**

| Characteristic | Category | Inadequate | | Marginal | | Adequate | | p-value |
|---|---|---|---|---|---|---|---|---|
| | | n | % | n | % | n | % | |
| Gender | male | 793 | 33.0% | 621 | 25.9% | 986 | 41.1% | < .001 |
| | female | 1,483 | 45.4% | 824 | 25.3% | 956 | 29.3% | |
| Age, years | 19–49 | 44 | 10.0% | 80 | 18.2% | 315 | 71.8% | < .001 |
| | 50–59 | 115 | 15.4% | 168 | 22.5% | 463 | 62.1% | |
| | 60–69 | 496 | 28.4% | 540 | 30.9% | 711 | 40.7% | |
| | 70–79 | 1,052 | 54.4% | 500 | 25.8% | 383 | 19.8% | |
| | ≥80 | 569 | 71.5% | 157 | 19.7% | 70 | 8.8% | |
| Educational level | no or primary education | 1,397 | 63.9% | 528 | 24.2% | 260 | 11.9% | < .001 |
| | middle school | 402 | 38.9% | 326 | 31.6% | 305 | 29.5% | |
| | high school | 373 | 23.4% | 434 | 27.2% | 788 | 49.4% | |
| | college or higher | 104 | 12.2% | 157 | 18.5% | 589 | 69.3% | |
| Marital status | currently married | 1,418 | 35.3% | 1,070 | 26.7% | 1,526 | 38.0% | < .001 |
| | formerly married | 810 | 55.2% | 346 | 23.6% | 311 | 21.2% | |
| | single | 48 | 26.4% | 29 | 15.9% | 105 | 57.7% | |
| Employment status | employed | 534 | 29.1% | 458 | 25.0% | 840 | 45.9% | < .001 |
| | self-employed | 385 | 39.5% | 243 | 24.9% | 347 | 35.6% | |
| | unemployed | 1,357 | 47.5% | 744 | 26.1% | 755 | 26.4% | |
| Household income quartile | 1 (lowest) | 855 | 60.5% | 317 | 22.4% | 241 | 17.1% | < .001 |
| | 2 | 670 | 47.4% | 385 | 27.2% | 359 | 25.4% | |
| | 3 | 451 | 31.9% | 404 | 28.6% | 559 | 39.5% | |
| | 4 (highest) | 294 | 20.8% | 337 | 23.8% | 782 | 55.3% | |
| Disability | yes | 325 | 50.9% | 177 | 27.7% | 136 | 21.3% | < .001 |
| | no | 1,951 | 38.8% | 1,268 | 25.2% | 1,806 | 35.9% | |
| No. of chronic conditions | 1 | 671 | 28.1% | 582 | 24.4% | 1,135 | 47.5% | < .001 |
| | 2 | 706 | 41.8% | 465 | 27.5% | 518 | 30.7% | |
| | ≥3 | 899 | 56.7% | 398 | 25.1% | 289 | 18.2% | |

p-values were obtained by using the Rao-Scott chi-square test.

dimensions is highest for inadequate literacy and lowest for adequate literacy (Table 3). The mean EQ-5D index (± SE) is 0.864 (± 0.004) for inadequate, 0.917 (± 0.003) for marginal, and 0.944 (± 0.002) for adequate health literacy levels.

The results of regression analysis show that EQ-5D index is positively associated with marginal (B = 0.018, $p<0.001$) and adequate health literacy levels (B = 0.017, $p<0.001$) compared

**Table 3. Proportion of participants reporting problems in EQ-5D dimensions and mean EQ-5D index by health literacy levels.**

| | Inadequate (N = 2,264) | | Marginal (N = 1,437) | | Adequate (N = 1,940) | | |
|---|---|---|---|---|---|---|---|
| EQ-5D dimension | n | % | n | % | n | % | p-value |
| Mobility | 999 | 44.1 | 339 | 23.6 | 254 | 13.1 | < .001 |
| Self-care | 339 | 15.0 | 84 | 5.8 | 65 | 3.4 | < .001 |
| Usual activities | 722 | 31.9 | 225 | 15.7 | 161 | 8.3 | < .001 |
| Pain/discomfort | 1,378 | 60.9 | 712 | 49.5 | 708 | 36.5 | < .001 |
| Anxiety/depression | 468 | 20.7 | 244 | 17.0 | 213 | 11.0 | < .001 |
| Mean EQ-5D index (± SE) | 0.863 (± 0.004) | | 0.917 (± 0.003) | | 0.944 (± 0.002) | | < .001 |

p-values for EQ-5D dimensions were obtained by using the Rao-Scott chi-square test and the p-value for mean EQ-5D index by using the ANOVA.

**Table 4. Results of multiple regression analyses on EQ-5D index.**

| Characteristic (reference) | Category | Estimate | SE | p-value |
|---|---|---|---|---|
| Gender (female) | male | 0.009 | 0.003 | < .01 |
| Age (≥80), years | 19–49 | 0.050 | 0.007 | < .001 |
| | 50–59 | 0.047 | 0.006 | < .001 |
| | 60–69 | 0.057 | 0.005 | < .001 |
| | 70–79 | 0.046 | 0.005 | < .001 |
| Education level (no or primary) | middle school | 0.015 | 0.004 | < .001 |
| | high school | 0.030 | 0.004 | < .001 |
| | college or higher | 0.036 | 0.005 | < .001 |
| Marital status (single) | currently married | 0.010 | 0.006 | 0.119 |
| | formerly married | -0.002 | 0.007 | 0.800 |
| Employment status (unemployed) | employed | 0.021 | 0.003 | < .001 |
| | self-employed | 0.027 | 0.004 | < .001 |
| Household income quartile (1) | 2 | 0.012 | 0.004 | < .01 |
| | 3 | 0.022 | 0.005 | < .001 |
| | 4 (highest) | 0.020 | 0.005 | < .001 |
| Disability (yes) | no | 0.071 | 0.005 | < .001 |
| No. of chronic conditions (≥3) | 1 | 0.052 | 0.004 | < .001 |
| | 2 | 0.032 | 0.004 | < .001 |
| Health literacy (inadequate) | marginal | 0.018 | 0.004 | < .001 |
| | adequate | 0.017 | 0.004 | < .001 |
| Constant | | 0.698 | 0.009 | < .001 |
| No. of observations | | | 5,632 | |
| R-square | | | 0.282 | |

to inadequate literacy (Table 4). The model accounts for 28.2% of variance in the EQ-5D index. Individuals with adequate literacy (OR = 1.5; 95% CI, 1.2–1.8; $p<0.001$) and those with marginal literacy (OR = 1.5; 95% CI, 1.2–2.0; $p<0.001$) are more likely to report no problem with mobility than those with inadequate literacy (Table 5). Adequate (OR = 1.6; 95% CI, 1.0–2.3; $p<0.05$) and marginal literacy (OR = 1.6; 95% CI, 1.1–2.2; $p<0.05$) increase the odds of having no problem with self-care. Individuals with adequate literacy (OR = 1.6; 95% CI, 1.2–2.2; $p<0.01$) and those with marginal literacy (OR = 1.4; 95% CI, 1.1–1.9; $p<0.01$) are more likely to report no problem with usual activities than those with inadequate literacy. Adequate health literacy is associated with an increased likelihood of reporting no problem with anxiety/depression (OR = 1.4; 95% CI, 1.1–1.9; $p<0.05$). Health literacy is not associated with the likelihood of having problems with pain and discomfort.

In sum, these findings are worth reiterating with respect to the hypotheses formulated at the inception of this study. First, health literacy is positively associated with HRQoL in patients with chronic conditions. Not only adequate but also marginal health literacy are linked to improved HRQoL. Second, health literacy has a significant impact on mobility, self-care, usual activities, and anxiety/depression but not on pain/discomfort.

## Discussion

This study shows that a combined total of 55.4% of Korean adults with chronic conditions have limited health literacy (30.8% for inadequate and 24.6% for marginal). This prevalence is more or less comparable to that in the general population of the U.S. [6], European countries [5,19,43], and Southeast Asian countries [44]. In the study presented here, adequate health

**Table 5. Results of logistic regression analyses for the five dimensions of EQ-5D.**

| Characteristic (reference) | Category | Mobility | | | | Self-care | | | | Usual activities | | | | Pain/discomfort | | | | Anxiety/depression | | | |
|---|---|---|---|---|---|---|---|---|---|---|---|---|---|---|---|---|---|---|---|---|---|
| | | OR | 95% CI | | p | OR | 95% CI | | p | OR | 95% CI | | p | OR | 95% CI | | p | OR | 95% CI | | p |
| Gender (female) | male | 1.2 | 1.0 | 1.5 | 0.051 | 0.6 | 0.5 | 0.9 | < .01 | 1.1 | 0.9 | 1.6 | 0.34 | 1.6 | 1.4 | 2.0 | < .001 | 1.2 | 1.0 | 1.5 | 0.087 |
| Age (≥80), years | 19–49 | 3.5 | 1.8 | 6.6 | < .001 | 2.6 | 1.0 | 6.5 | < .05 | 2.6 | 1.3 | 5.4 | < .01 | 1.9 | 1.3 | 3.0 | < .01 | 0.9 | 0.5 | 1.5 | 0.589 |
| | 50–59 | 2.8 | 1.9 | 4.2 | < .001 | 3.2 | 1.7 | 6.0 | < .001 | 3.0 | 1.9 | 4.9 | < .001 | 1.8 | 1.3 | 2.5 | < .001 | 0.9 | 0.6 | 1.3 | 0.531 |
| | 60–69 | 2.8 | 2.1 | 3.8 | < .001 | 3.5 | 2.4 | 5.3 | < .001 | 3.4 | 2.5 | 4.6 | < .001 | 2.0 | 1.6 | 2.7 | < .001 | 1.0 | 0.8 | 1.4 | 0.803 |
| | 70–79 | 1.9 | 1.5 | 2.5 | < .001 | 2.3 | 1.7 | 3.2 | < .001 | 2.1 | 1.7 | 2.8 | < .001 | 1.8 | 1.4 | 2.3 | < .001 | 1.2 | 1.0 | 1.6 | 0.108 |
| Education level (no or primary) | middle school | 1.4 | 1.1 | 1.8 | < .05 | 1.0 | 0.7 | 1.5 | 0.974 | 1.2 | 0.9 | 1.5 | 0.304 | 1.2 | 1.0 | 1.5 | 0.111 | 1.2 | 0.9 | 1.5 | 0.313 |
| | high school | 1.8 | 1.4 | 2.4 | < .001 | 1.3 | 0.9 | 2.0 | 0.180 | 1.6 | 1.2 | 2.2 | < .01 | 1.7 | 1.4 | 2.1 | < .001 | 1.4 | 1.0 | 1.8 | < .05 |
| | college or higher | 2.4 | 1.7 | 3.5 | < .001 | 1.5 | 0.8 | 2.7 | 0.208 | 2.5 | 1.6 | 3.9 | < .001 | 2.1 | 1.6 | 2.9 | < .001 | 1.5 | 1.0 | 2.2 | 0.058 |
| Marital status (single) | currently married | 1.5 | 0.8 | 2.8 | 0.246 | 1.1 | 0.5 | 2.4 | 0.863 | 1.6 | 0.8 | 3.1 | 0.199 | 1.1 | 0.7 | 1.8 | 0.600 | 1.4 | 0.8 | 2.3 | 0.198 |
| | formerly married | 1.2 | 0.6 | 2.2 | 0.665 | 0.6 | 0.3 | 1.5 | 0.300 | 1.3 | 0.6 | 2.5 | 0.504 | 0.9 | 0.6 | 1.4 | 0.619 | 1.1 | 0.7 | 1.9 | 0.662 |
| Employment status (unemployed) | employed | 1.7 | 1.3 | 2.1 | < .001 | 2.4 | 1.6 | 3.5 | < .001 | 1.5 | 1.2 | 2.0 | < .01 | 1.3 | 1.0 | 1.5 | < .05 | 1.4 | 1.1 | 1.8 | < .01 |
| | self-employed | 1.9 | 1.4 | 2.5 | < .001 | 2.9 | 1.8 | 4.5 | < .001 | 2.2 | 1.5 | 3.1 | < .001 | 1.3 | 1.0 | 1.7 | < .05 | 1.5 | 1.1 | 2.0 | < .05 |
| Household income quartile (1) | 2 | 1.2 | 0.9 | 1.5 | 0.131 | 1.0 | 0.7 | 1.4 | 0.933 | 1.1 | 0.8 | 1.4 | 0.526 | 1.2 | 1.0 | 1.5 | 0.100 | 1.2 | 0.9 | 1.6 | 0.121 |
| | 3 | 1.6 | 1.2 | 2.1 | < .001 | 1.7 | 1.1 | 2.5 | < .05 | 1.7 | 1.2 | 2.3 | < .01 | 1.3 | 1.0 | 1.7 | < .05 | 1.4 | 1.0 | 1.9 | < .01 |
| | 4 | 1.5 | 1.1 | 2.0 | < .05 | 1.7 | 1.0 | 3.0 | < .05 | 1.6 | 1.1 | 2.3 | < .05 | 1.3 | 1.0 | 1.8 | < .05 | 1.7 | 1.2 | 2.4 | < .01 |
| Disability (yes) | no | 3.6 | 2.7 | 4.7 | < .001 | 3.9 | 2.8 | 5.4 | < .001 | 3.4 | 2.6 | 4.6 | < .001 | 2.1 | 1.6 | 2.7 | < .001 | 1.4 | 1.1 | 1.9 | < .05 |
| No. of chronic conditions (≥3) | 1 | 3.0 | 2.4 | 3.8 | < .001 | 2.7 | 1.9 | 3.9 | < .001 | 2.8 | 2.1 | 3.6 | < .001 | 2.4 | 1.9 | 3.0 | < .001 | 1.6 | 1.2 | 2.1 | < .001 |
| | 2 | 1.6 | 1.3 | 2.0 | < .001 | 1.5 | 1.1 | 2.0 | < .05 | 1.6 | 1.3 | 2.0 | < .001 | 1.4 | 1.2 | 1.8 | < .001 | 1.3 | 1.1 | 1.7 | < .05 |
| Health literacy (inadequate) | marginal | 1.5 | 1.2 | 1.8 | < .001 | 1.6 | 1.1 | 2.2 | < .05 | 1.4 | 1.1 | 1.8 | < .01 | 1.0 | 0.8 | 1.2 | 0.751 | 1.0 | 0.8 | 1.2 | 0.732 |
| | adequate | 1.5 | 1.2 | 2.0 | < .001 | 1.6 | 1.0 | 2.3 | < .05 | 1.6 | 1.2 | 2.2 | < .01 | 1.0 | 0.8 | 1.3 | 0.916 | 1.4 | 1.1 | 1.9 | < .05 |

Abbreviations: OR, Odds ratio; CI, Confidence interval.

literacy is positively associated with HRQoL among patients with chronic diseases, confirming the first hypothesis established at the outset and consistent with the findings of earlier studies [14,29,30].

The observed association may be explained by several mechanisms. First of all, health literacy can affect patients' knowledge and belief about diseases that may be helpful to the management of chronic conditions [45]. For example, chronic obstructive pulmonary disease (COPD) patients with low health literacy are less likely to believe that they will always have the chronic condition and more likely to express concerns about medications [46], which could affect their adherence to treatment. Second, health literacy influences heath care utilization that may be necessary to appropriately manage chronic conditions [18,19]. For example, patients with chronic low back pain that have adequate health literacy are more likely to use medications and see a specialist than those with limited literacy. Medication adherence, in particular, appears to be influenced by health literacy [47]. This may partly by low literacy adversely influencing adult patients' ability to correctly interpret warning labels for prescription drugs [48]. Third, adequate health literacy could positively affect patients' engagement and empowerment in their self-care [19]. Patients with limited health literacy are more likely to experience difficulty using health materials received from health professionals, thereby negatively affecting their self-management and decision-making [49]. Adequate health literacy can be particularly important for patients with complex chronic diseases who need to be actively involved in their

own care [50]. Diabetes patients with limited literacy have a higher risk of physical inactivity and unhealthy diet [51].

The proportion of participants reporting moderate or severe problems is highest in people with inadequate literacy and lowest in those with adequate literacy in all five dimensions of EQ-5D. However, after adjustments for covariates, adequate health literacy is associated with the odds of reporting no problems with self-care, mobility, usual activities, and anxiety/depression only. The existing literature supports the link between these specific dimensions of HRQoL and health literacy, based on some demographic groups [52,53] and a few common chronic conditions [54,55]. For example, the significant influence of health literacy on self-care behaviors has been documented in patients with diabetes [54], hypertension [55], and heart failure [56]. Adequate health literacy is linked to lower disease activity and improved physical functioning among patients with rheumatic diseases [57]. A number of studies find that better health literacy is associated with lower risks for anxiety and depressive symptoms [52,58–60]. Consistent with the findings, this current study shows that adults with adequate literacy are more likely to report no problems with anxiety/depression than those with inadequate literacy. It is plausible that adequate literacy may help patients cope with mental conditions associated with chronic diseases.

Health literacy has been shown to be inversely associated with pain intensity among patients with musculoskeletal pain [61] and those with chronic pain [62]. One possible explanation for the relationship points to the role of health literacy in better pain management [62]. Limited health literacy adversely influences knowledge about overall pain medication and non-medication modes of pain management [63]. However, this current study does not support the link between health literacy and the pain/discomfort dimension of EQ-5D. The contradictory finding may have to do with the choice of the study population; this present study is not limited to patients suffering from pain but based on a sample of adults with any of chronic conditions.

This study shows that health literacy is positively associated with the EQ-5D index among adults with chronic diseases. In addition, adults with adequate health literacy are more likely to report no problems with mobility, self-care, usual activities, and anxiety/depression, but not with pain/discomfort, which confirms the hypotheses of this study. These findings have the following implications. First, inadequate health literacy is so prevalent in adults with chronic diseases that across-the-board public health interventions to improve health literacy is warranted. Efforts to improve health literacy among adults with chronic conditions can be particularly important since patient-reported HRQoL is deteriorated in chronically ill patients. Second, the development of health literacy interventions should prioritize vulnerable subpopulations based on sociodemographic and health-related characteristics for improved efficiency.

This study has the following limitations. First, this analysis of cross-sectional data cannot establish a causal relationship between health literacy and HRQoL. Second, this study is based on self-reported data on chronic conditions and other characteristics and therefore may be subject to recall and reporting biases. Third, a substantial ceiling effect is reported for rating of EQ-5D by patients with chronic conditions [26]. Future research should explore the possibility of using other measures of HRQoL.

## Conclusion

This population-based study shows that limited health literacy is a prevalent problem in adults with chronic diseases in Korea. Adequate health literacy is associated with better HRQoL. Health literacy has a differential impact across dimensions of HRQoL. Adequate health literacy is a protective factor for four dimensions of EQ-5D (mobility, self-care, usual activities, and

anxiety/depression). Considering the high prevalence and its negative impact on HRQoL, limited health literacy should be recognized as a public health concern among adults with chronic diseases.

## Author Contributions

**Conceptualization:** Inmyung Song.

**Data curation:** Inmyung Song.

**Formal analysis:** Inmyung Song.

**Investigation:** Inmyung Song.

**Visualization:** Inmyung Song.

**Writing – original draft:** Inmyung Song.

**Writing – review & editing:** Inmyung Song.

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
