## [Decision Letter · Decision Letter 0]

2 Feb 2024

PONE-D-23-40500Relationship between health literacy and health-related quality of life in Korean adults with chronic diseasesPLOS ONE

Dear Dr. Song,

Thank you for submitting your manuscript to PLOS ONE. After careful consideration, we feel that it has merit but does not fully meet PLOS ONE’s publication criteria as it currently stands. Therefore, we invite you to submit a revised version of the manuscript that addresses the points raised during the review process.

As it is very difficult to get reviewers, I have performed one of the two reviews myself. I am reviewer 2. Except for two comments, I also agree with the comments of reviewer 1. One comment with which I do not agree is the first reviewer’s comment regarding binomial logistic regression. As far as I can see you have applied this kind of regression in exactly those cases in which it is necessary. A further comment with which I not completely agree is given in the first paragraph regarding the results. In contrast to reviewer 1, I do not think that the description is too brief. However, I share with reviewer 1 a feeling of not being completely satisfied with this description. My problem is caused by the fact that you mainly repeat the information given in the tables. I would like to have a description that focusses more on the more essential aspects in the data, i.e., that you already elaborate and highlight what is especially important. This might be similar with what reviewer 1 meant with: ‘A more detailed description in terms of striking or special results would be useful. Especially those that are to be discussed.’

I must also give two comments that I forgot in my review. The first comment regards the fact that you use the Rao-Scott chi-square test for testing differences between categorial variables. The usual approach is applying that chi-square test that has originally been presented by Pearson and that is presented in textbooks as the classical approach for analyzing cross-tables. Please, explain what the Rao-Scott chi-square test distinguishes from the Pearson chi-square and why you think that choosing this statistic is more appropriate for your problem. If there is no special reason for choosing the Rao-Scott chi-square test I would like you to rerun the corresponding analyses with Pearson’s chi-square. This statistic has the simplest and clearest meaning. The second additional comment regards the footnote of table 3. This footnote is: ‘p-values were obtained by using the Rao-Scott chi-square test and the ANOVA.’ This makes no sense because you report only one p-value per test and an ANOVA would not be appropriate in this context. Please report the one and only statistical procedure you have actually applied.

We look forward to receiving your revised manuscript.

Kind regards,

Uwe Konerding

Academic Editor

PLOS ONE

Journal Requirements:

2. In the online submission form, you indicated that your data is available only on request from a third party. Please note that your Data Availability Statement is currently missing [the name of the third party contact or institution / contact details for the third party, such as an email address or a link to where data requests can be made]. Please update your statement with the missing information. 

Reviewers' comments:

Reviewer's Responses to Questions

**Comments to the Author**

1. Is the manuscript technically sound, and do the data support the conclusions?

Reviewer #1: Yes

Reviewer #2: Yes

2. Has the statistical analysis been performed appropriately and rigorously? 

Reviewer #1: Yes

Reviewer #2: Yes

3. Have the authors made all data underlying the findings in their manuscript fully available?

Reviewer #1: Yes

Reviewer #2: No

4. Is the manuscript presented in an intelligible fashion and written in standard English?

Reviewer #1: Yes

Reviewer #2: Yes

5. Review Comments to the Author

Reviewer #1: Dear author,

Many thanks for the exciting manuscript. The presentation is largely solid. Below are a few points that could supplement the manuscript

In general, the work of the Asian Health Literacy Association (AHLA), of which South Korea is also a member, could be discussed in a few sentences. This would be of interest to the international readership. For articles on national aspects of health literacy, it would be helpful to add national circumstances (education system, socioeconomics, utilisation of the healthcare system). This does not have to be described in detail. But appropriate sources would be helpful.

Introduction

The work of the AHLA could be mentioned in the introduction. In addition, something should be mentioned about the Korean education system in the context of health literacy so that the international readership can categorise it.

Otherwise, the points listed above would be useful in the introduction.

Methods:

Justify the use of HLS-EU 16! At the time of the study, the HLS-SF12 had already been successfully validated in Asia [Duong et al., 2019].

The choice of covariates should be justified. Most of them are plausible. However, for example, the severity of an illness rather than the number of illnesses seems to me to be more important. Or number of household members seems more logical to me than marital status. Please explain the choice of covariates.

When carrying out the logistic regression analysis, the question arises as to why binomial logistic regression was not used for dichotomous (binary) dependent variables. Please explain! In many places, the binomial variant would have been more meaningful.

Results

The results are presented in a solid table. However, the description of the results is rather brief. A more detailed description in terms of striking or special results would be useful. Especially those that are to be discussed.

With regard to the hypotheses mentioned in the introduction, the presentation of the results should also be expanded in this respect.

Discussion

The discussion is sound and the limitations are clearly stated. However, the concept of hypothesis could also be included here.

Conclusion

Conclusion can be drawn!

see method:

Duong TV, Aringazina A, Kayupova G, Nurjanah, Pham TV, Pham KM, Truong TQ, Nguyen KT, Oo WM, Su TT, Majid HA, Sørensen K, Lin IF, Chang Y, Yang SH, Chang PWS. Development and Validation of a New Short-Form Health Literacy Instrument (HLS-SF12) for the General Public in Six Asian Countries. Health Lit Res Pract. 2019 Apr 10;3(2):e91-e102. doi: 10.3928/24748307-20190225-01.]

Reviewer #2: Review regarding the manuscript ‘Relationship between health literacy and health-related quality of life in Korean adults with chronic diseases’

A study regarding the relationship between health literacy and health-related quality of life in Korean adults with chronic diseases is presented in the manuscript. The study is performed with cross-sectional data taken from the Korea Health Panel Survey from 2021. The topic of the manuscript is interesting. The analyses are methodologically sound, and the different consideration are well ordered. However, there are several minor flaws. Most of them refer to the language. I have tried to address all of these flaws, including those regarding the language. However, I am not a native speaker of the English language. Therefore, I recommend having the manuscript being checked by a native speaker. Should this person make different propositions than I make, please discuss this with the native speaker and choose that solution that the native speaker and you finally prefer. In the following text, you find the numerous minor flaws that should be corrected before publication.

Lines 21-22

‘This aims to examine the relationship between health literacy and HRQoL among Korean adults with chronic conditions. This cross-sectional study was based on the Korea Health Panel Survey in 2021.’

I suggest replacing this by

‘In this contribution, the relationship between health literacy and HRQoL in Korean adults with chronic conditions is examined using data of the cross-sectional Korea Health Panel Survey from 2021.’

Line 25 and further text

I suggest replacing

‘EQ-5D index score’

by

‘EQ-5D index’

An index is a score. Therefore, ‘EQ-5D index score’ is one word too much.

Lines 26 to 28

‘Multiple logistic regression models were implemented for domain-specific outcomes. In the weighted sample, 30.8%, 24.6%, and 44.6% had inadequate, marginal, and adequate levels of health literacy, respectively.’

A reader who has not read your manuscript cannot know what you meant by ‘domain-specific’ and, even a reader who has read you manuscript does not know what you mean by weighted by ‘weighted’? (See below)

Please revise the corresponding parts of the main text according to my suggestions and revise the abstract correspondingly.

Lines 28 to 30

‘EQ-5D index scores were positively associated with marginal (B=0.018, p<0.001) and adequate literacy (B=0.017, p<0.001) compared to inadequate literacy.’

I suggest writing

‘The EQ-5D index increases with marginal (B=0.018, p<0.001) and adequate literacy (B=0.017, p<0.001) compared to inadequate literacy.’

Lines 34 to 35

‘Inadequate health literacy is prevalent among Korean adults with chronic diseases.’

Should be reported earlier.

*****

Lines 47 to 48

‘Nearly a half of adults in the U.S. have limited health literacy according a systematic review [6].’

I suggest replacing this by

‘According to a systematic review [6], nearly a half of adults in the U.S. have limited health literacy.’

Line 48

I suggest replacing ‘meaning’ by ‘which means that’.

Lines 49 to 51

I suggest replacing

‘Nearly one in two adults aged 19 years and over in Korea have limited health literacy to make informed decision regarding their own health [8].’

By

‘In Korea, nearly one in two adults aged 19 years and over has limited health [8].’ The term ‘to make informed decision regarding their own health’ is not helpful in this sentence. The remaining changes regard the language.’

Line 57

Please replace ‘component to effective treatment’ by ‘component of effective treatment’.

Line 59

Please replace ‘negatively can’ by ‘can negatively’.

Line 65

With regard to the language, you should replace ‘Reflective of’ by ‘Reflecting’ or ‘Being reflective of’. However, there is still a further problem in the sentence. HRQoL-indices as the EQ-5D-index do not reflect the individual experience. They reflect societal values.

Line 65 to 77

Please report findings in present tense. They are usually meant to be generalizable over time.

Line 70 to 71

Replace ‘Type 2 diabetes is a chronic condition that garnered particular attention in relation to health literacy [22–24]’ by ‘A chronic condition that garnered particular attention in relation to health literacy is type 2 diabetes [22–24]’.

Line 77

Please replace ‘Also’ by ‘Moreover’ or ‘Further’ or ‘Furthermore’.

Line 78

Please omit ‘using survey data based on a nationally representative large sample of the population’. Inserting this phrase makes the sentence very difficult to understand. Please give this information in the first sentence of the methods part.

Lines 83 and 84

Please formulate the hypotheses in present tense (see above).

Line 91

‘city vs. province’

Why ‘vs’? I cannot be certain about what you want to say, but I guess the formulation ‘or, respectively,’ accords more to what you might want to express.

Line 92

The word ‘dong’ belongs to American slang and means, with absolute certainty, something that you do not want to talk about in this manuscript. The word ‘eup’ is not an English word at all. Please use English words. Moreover, my comment regarding ‘vs’ also applies to this line.

Line 95

I would leave out ‘presenting a rare opportunity to examine the hypotheses posited above’.

Line 111 to 119

I am not certain how this paragraph is meant. Is it a list of different health literacy measurement instruments included in the survey or is it consideration of the development of these measurement over time? If the first is true, you should make this clear. For example, you could write ‘KHPS contained several questionnaires addressing health literacy. These questionnaires are….’. If the second is true, this part belongs into the introduction.

Line 120 to 128

Please give first the general description of the HLS-EU-Q16 and then the report of the cultural adaptation.

Lines 131 to 142

You should report that each dimension of the EQ-5D is addressed by exactly one item, and you should replace the formulation ‘domain specific responses’ by the formulation ‘responses to the individual EQ-5D items’ throughout the complete manuscript. This will make your presentation easier to understand.

Sub-chapter ‘Covariates’

Please, tell the reader purpose for which you want to control for confounding variables and why you believe that the variables you have included as possible confounders serve this purpose.

Line 149

'…classified as ≤primary, middle school, high school, and ≥college’

What do the symbols ‘≤’ and ‘≥’ mean in this context? I guess that they should be omitted.

Line 150

Please replace ‘ever married’ by ‘formerly been married’.

Line 151

I suggest using the denominations ‘employed’, ‘self-employed’ and ‘un-employed’ for categorizing employment status.

Sub-chapter ‘Statistical analysis’

Past-tense sounds better and is more often used in the analysis sub-chapter.

Line 164

As far as I can infer from your text, you actually apply multivariate and not multivariable regression. As far as I know, multivariable regression is a regression with more than one dependent variable.

Lines 169 to 170

‘All analyses adjusted for sampling weights that are used in the complex sampling design of the 2021 KHPS.’

I have no idea what is done here. Please explain in the text. What is weighted by what? To which purpose is this weighting performed? How is the weighting integrated in the analyses?

Chapter ‘results’

I would use present tense throughout this chapter.

Line 175

What do you want to say with ‘In all’ in this context?

Please replace ‘received’ by ‘had’.

Line 182

Please place the adverb after the verb.

Chapter ‘Discussion’

I also would use present tense in this chapter. Moreover, this chapter is very long detailed. Most readers would be thankful if you shortened the discussion to a half of the present length and focus on the more general aspects.

Line 219

Please replace ‘this present study’ by ‘the study presented here’.

Lines 220 to 222

‘In particular, multi-morbidity appears to be an influential factor to the prevalence of inadequate health literacy, which increased in patients with a greater number of chronic conditions.’

You have cross-sectional data. Therefore, you cannot infer from your data whether a relationship between two variables results from one variable influencing the other, neither can you, in case of an influence, infer from the data, which of the two variables is the cause and which the effect. Youd mention something like this later as limitation. Accordingly, you should be cautious with using words as influence in this context. Of course, you can speculate about causal relationships as far as additional knowledge regarding the object of investigating allows for. However, given my knowledge regarding the object of investigation I see only a few mechanisms in which multi-morbidity could influence health literacy. To be specific, I do think that dementia reduces health literacy. However, I actually assume that health literacy influences multi-morbidity. To be specific, the higher the health literacy is the more people behave in a health-promoting way and, consequently, the less people suffer from multi-morbidity. In the further text, you develop similar thoughts. You should make the text in lines 220 to 222 consistent with this line of thinking.

Lines 238 to 240

‘This may be in part because low literacy adversely influenced adult patients’ ability to correctly interpret warning labels for prescription drugs [40].’

I suggest

‘This may partly be caused by low literacy adversely influencing adult patients’ ability to correctly interpret warning labels for prescription drugs [40].’

Lines 252 to 254

‘However, after adjustments for covariates, adequate and marginal health literacy were associated with the odds of reporting no problems with self-care, mobility, and usual activities only.’

I would restrict to the category of adequate health literacy and, accordingly, include anxiety/depression into the set of dimensions associated with health literacy.

Line 254

If you follow my suggestion to replace ‘domain specific responses’ by the formulation ‘responses to the individual EQ-5D items’ then you should replace in line 254 ‘domains’ by ‘dimensions’.

Chapter ‘Conclusion’ please also in present tense.

6. PLOS authors have the option to publish the peer review history of their article (what does this mean?). If published, this will include your full peer review and any attached files.

Reviewer #1: No

Reviewer #2: **Yes: **Uwe Konerding

---

## [Decision Letter · Decision Letter 1]

26 Mar 2024

Relationship between health literacy and health-related quality of life in Korean adults with chronic diseases

PONE-D-23-40500R1

Dear Dr. Song,

We’re pleased to inform you that your manuscript has been judged scientifically suitable for publication and will be formally accepted for publication once it meets all outstanding technical requirements.

Kind regards,

Uwe Konerding

Academic Editor

PLOS ONE

Additional Editor Comments (optional):

Reviewers' comments:

Reviewer's Responses to Questions

**Comments to the Author**

1. If the authors have adequately addressed your comments raised in a previous round of review and you feel that this manuscript is now acceptable for publication, you may indicate that here to bypass the “Comments to the Author” section, enter your conflict of interest statement in the “Confidential to Editor” section, and submit your "Accept" recommendation.

Reviewer #1: All comments have been addressed

Reviewer #2: All comments have been addressed

2. Is the manuscript technically sound, and do the data support the conclusions?

Reviewer #1: Yes

Reviewer #2: Yes

3. Has the statistical analysis been performed appropriately and rigorously? 

Reviewer #1: Yes

Reviewer #2: Yes

4. Have the authors made all data underlying the findings in their manuscript fully available?

Reviewer #1: Yes

Reviewer #2: No

5. Is the manuscript presented in an intelligible fashion and written in standard English?

Reviewer #1: Yes

Reviewer #2: Yes

6. Review Comments to the Author

Reviewer #1: Dear Author,

Congratulations on the successful revision. All my comments and questions have been fully or sufficiently addressed and taken into account.

As far as the use of the HLS-EU12 is concerned, I suspect that due to a long planning and realisation period, the validation ran right into the implementation phase and therefore no further adjustments could be made.

I therefore recommend publication of the manuscript.

I wish you continued success!!!

Reviewer #2: You have addressed all of my comments. Now, the paper is fine. There are no objections to publication any longer.

7. PLOS authors have the option to publish the peer review history of their article (what does this mean?). If published, this will include your full peer review and any attached files.

Reviewer #1: No

Reviewer #2: No

---

## [Editor Report · Acceptance letter]

29 Mar 2024

PONE-D-23-40500R1 

PLOS ONE

Dear Dr. Song, 

I'm pleased to inform you that your manuscript has been deemed suitable for publication in PLOS ONE. Congratulations! Your manuscript is now being handed over to our production team.

Kind regards, 

on behalf of

Dr. Uwe Konerding 

Academic Editor

PLOS ONE